# Synthesis of Antibacterial Copper Oxide Nanoparticles by Pulsed Laser Ablation in Liquids: Potential Application against Foodborne Pathogens

**DOI:** 10.3390/nano13152206

**Published:** 2023-07-29

**Authors:** Tina Hesabizadeh, Kidon Sung, Miseon Park, Steven Foley, Angel Paredes, Stephen Blissett, Gregory Guisbiers

**Affiliations:** 1Department of Physics and Astronomy, University of Arkansas at Little Rock, 2801 South University Avenue, Little Rock, AR 72204, USA; txhesabizade@ualr.edu (T.H.); sxblissett@ualr.edu (S.B.); 2Division of Microbiology, National Center for Toxicological Research, US Food and Drug Administration, Jefferson, AR 72079, USA; kidon.sung@fda.hhs.gov (K.S.); miseon.park@fda.hhs.gov (M.P.); steven.foley@fda.hhs.gov (S.F.); 3NCTR-ORA Nanotechnology Core Facility, National Center for Toxicological Research, US Food and Drug Administration, Jefferson, AR 72079, USA; angel.paredes@fda.hhs.gov

**Keywords:** nanoparticles, copper oxide, pulsed laser ablation in liquids, antibacterial

## Abstract

Spherical copper oxide nanoparticles (CuO/Cu_2_O NPs) were synthesized by pulsed laser ablation in liquids (PLAL). The copper target was totally submerged in deionized (DI) water and irradiated by an infrared laser beam at 1064 nm for 30 min. The NPs were then characterized by dynamic light scattering (DLS) and atomic emission spectroscopy (AES) to determine their size distribution and concentration, respectively. The phases of copper oxide were identified by Raman spectroscopy. Then, the antibacterial activity of CuO/Cu_2_O NPs against foodborne pathogens, such as *Salmonella enterica* subsp. *enterica* serotype Typhimurium DT7, *Escherichia coli* O157:H7, *Shigella sonnei* ATCC 9290, *Yersinia enterocolitica* ATCC 27729, *Vibrio parahaemolyticus* ATCC 49398, *Bacillus cereus* ATCC 11778, and *Listeria monocytogenes* EGD, was tested. At a 3 ppm concentration, the CuO/Cu_2_O NPs exhibited an outstanding antimicrobial effect by killing most bacteria after 5 h incubation at 25 °C. Field emission scanning electron microscope (FESEM) confirmed that the CuO/Cu_2_O NPs destructed the bacterial cell wall.

## 1. Introduction

Foodborne pathogenic microorganisms (bacteria, fungi, and viruses) are microbes that contaminate food or drinks, and in the most extreme cases cause foodborne illnesses. This type of illness presents a major public health problem in the U.S. and throughout the world [1,2]. Indeed, the Centers for Disease Control and Prevention (CDC) estimates that 48 million Americans suffer from foodborne illness each year, resulting in 128,000 hospitalizations and 3,000 deaths annually [3,4]. In 2018, the estimated cost of foodborne illnesses in the US was about $17.6 billion [5]. To prevent contamination and reduce fatal issues, antibiotics have been used prophylactically and therapeutically. The problem is that the same antibiotics have been used systematically as growth promoters in the food industry, giving birth to antibiotic resistance. Consequently, antibiotic resistance of foodborne pathogens is an inevitable side-effect of the indiscriminate use of antibiotics, as antibiotic-resistant bacteria can be transmitted to humans through the food chain [6]. Therefore, foodborne pathogens are responsible of creating the “antibiotic-resistance” problem, which causes more than 2.8 million infections and costs USD 35 billion in productivity losses each year in the US [7].

A potential solution may come from nanotechnologies. Indeed, nanoparticles (NPs) have received considerable interest because of their huge surface-to-volume ratio [8,9,10,11] leading to various applications in agriculture [12] and medicine [13]. Among all the NPs, copper oxide (CuO/Cu_2_O) NPs are one of the most popular because they are inexpensive and abundant compared to other antibacterial NPs, such as silver or gold [4,14,15]. Furthermore, CuO/Cu_2_O are chemically stable, and have a long shelf life. They are used in drug delivery carriers, antimicrobial agents, antioxidants, anticancer agents, and therapeutics in the biomedical area [16,17]. Specifically, these NPs have shown antimicrobial activity against a wide range of microorganisms including bacteria, fungi, and viruses [18,19] by disrupting cell walls and causing cell damage [20]. Indeed, CuO/Cu_2_O NPs interact with sulfhydryl (-SH) groups and produce reactive oxygen species (ROS) that ultimately cause irreversible damage to the cells of pathogenic bacteria.

The novelty of this work is to synthesize antibacterial CuO/Cu_2_O NPs, by using a synthesis protocol called “pulsed laser ablation in liquids” (PLAL) [21,22], and test them against foodborne pathogens. This simple process allows for the synthesis of CuO/Cu_2_O NPs directly into solution without the addition of surfactants, consequently exhibiting a totally clean surface when interacting with the bacteria [23]. To avoid agglomeration, the electrical charge at the NP surface is used to promote electrostatic repulsion between NPs. The paper is organized as follows; Section 2 describes the experimental protocol, Section 3 presents the results, Section 4 discusses the results, and Section 5 summarizes the conclusions.

## 2. Experimental Section

### 2.1. Synthesis of the CuO/Cu_2_O NPs

A Nd:YAG laser from Electro Scientific Industries was used to synthesize spherical copper oxide nanoparticles. The laser beam was emitted in the infrared region of the spectrum at 1064 nm. The beam was reflected at a 90 degrees angle by a gold coated mirror and focused onto the copper target with a biconvex lens having a focal length of ~83mm. The copper target was made of 15 copper beads (99.99% pure, Sigma Aldrich, St. Louis, MO, USA) totally submerged into a 25 mL rounded single-neck flask filled with 3 mL of deionized (DI) water. The height of DI water above the copper targets was around 10 mm. The repetition rate used for this synthesis was set at 5.1 kHz [23]. The energy per pulse at 5.1 kHz was around ~2 mJ/pulse. The beam’s spot size on the target was measured by scanning electron microscopy (SEM) to be around ~50 μm. Consequently, the fluence was calculated to be around ~100 J/cm^2^ at 5.1 kHz. The irradiation lasted for 30 min.

### 2.2. Physico-Chemical Characterization of CuO/Cu_2_O NPs

The Raman spectrometer used in this investigation was a micro-Raman LabRam 800 from Horiba (Irvine, CA, USA). The Dynamic Light Scattering equipment was a NanoBrook 90PlusZeta from Brookhaven Instruments (Holtsville, NY, USA), operating at 25 degrees Celsius. The Scanning Electron Microscope (SEM) used to characterize the CuO/Cu_2_O NPs was a JSM—7000F from JEOL (Peabody, MA, USA) that employs a Schottky-type field emission gun for the electron source, and was operating at 15 kV.

### 2.3. Antimicrobial Test of CuO/Cu_2_O NPs

A single bacterial colony of *Salmonella enterica* serotype Typhimurium DT7, *Escherichia coli* O157:H7, *Shigella sonnei* ATCC 9290, *Yersinia enterocolitica* ATCC 27729, *Vibrio parahaemolyticus* ATCC 49398, *Bacillus cereus* ATCC 11778, and *Listeria monocytogenes* EGD was inoculated in 1 mL of brain heart infusion (BHI) broth (Thermo Scientific Remel, Lenexa, KS, USA) and grown in a shaking incubator (Eppendorf North America, Inc., Framingham, MA, USA) for 18 h at 37 °C. The culture was diluted 1:100 in fresh media and grown for 4 h. The cells were then harvested by centrifugation at 20,817× *g* at 4 °C and washed with phosphate-buffered saline (PBS) three times. The concentration of resuspended bacterial solution was adjusted to an optical density at 600 nm of 0.01 with CuO/Cu_2_O NPs and incubated at 25 °C for 5 h using the shaking incubator at 150 rpm. After incubation, samples were serially diluted with PBS and plated on the tryptic soy agar (TSA) plates. The agar plates were incubated overnight at 37 °C and the viable colony forming units (CFU/mL) were counted. A control group without CuO/Cu_2_O NPs was included in the experiment. All experiments were done in triplicate and the data are expressed as the mean ± standard error. Percent inhibition of the bacterial cells exposed to the NPs was calculated as follows: [(CFU/mL of Control cells − CFU/mL of Treated bacterial cells)/CFU/mL of Control cells)] × 100.

### 2.4. Observations of NP-Bacteria’s Interactions by Scanning Electron Microscopy

After CuO/Cu_2_O NPs treatment, the bacterial cells were washed with PBS and dehydrated using 15%, 30%, 50%, 70%, 80%, 90%, 95%, and 100% ethanol, performing each step for 10 min. Samples were immersed with a 1:2, 1:1, and 2:1 mixture of hexamethyldisilazane (HMDS, Sigma-Aldrich, St Louis, USA) and ethanol for 15 min each, and then pure HMDS for 20 min twice. A second wash of HMDS was allowed to dry in a fume hood overnight. Finally, samples were sputter-coated with gold (Denton Vacuum, Moorestown, NJ, USA), and images were visualized with the secondary electron detector (SE2) using a Zeiss-Merlin Field Emission-Scanning Electron Microscope (Carl Zeiss Microscopy, Thornwood, NY, USA) operating at 4.85 keV.

## 3. Results

CuO/Cu_2_O NPs were obtained by PLAL (Figure 1A). The presence of NPs was demonstrated by the Tyndall effect (Figure 1B). The phases of the NPs were identified by Raman spectroscopy (Figure 1C) and corresponded to CuO and Cu_2_O. As the target was static with respect to the laser beam, copper was ionized into Cu^+^ and Cu^2+^ ions which then recombined with O^2-^ ions, from the breakdown of water molecules, to form CuO and Cu_2_O. The growth mechanism was fully detailed in ref. [23]. The main difference between the synthesis performed by Hesabizadeh et al. [23] and the current one herein is that this one contains only a single irradiation step; the second irradiation step involving the electric field was omitted. The shape of the CuO/Cu_2_O NPs was confirmed to be spherical by scanning electron microscopy (SEM) (Figure 1D). Two main populations of particles were observed, one around ~100 nm and another one around ~1 micron, as demonstrated by Dynamic Light Scattering (DLS) (Figure 1E). As a reminder, the diameter measured by DLS is the hydrodynamic diameter of the nanoparticles and not their physical diameter [24]. In the intensity versus size distribution (Figure 1E), the intensity of peaks could be misleading, making the reader think that there is a huge population of particles displaying a size of around ~1 micron and a smaller population of particles around ~100 nm. But, particles scatter incident light proportional to the 6th power of their radii, meaning that larger particles will scatter more light than smaller ones. Another type of distribution may be obtained from DLS, and the number versus size distribution tells us that most of the particles in the colloid have a size below ~100 nm. By looking at the intensity and number distributions, the disappearance of the second peak in the number distribution for large particles around ~1 micron means that there is flocculation. Indeed, the zeta potential (ζ) was measured to be around −15 mV indicating a non-stable colloid (Figure 1F). Generally, a stable colloid displays a value larger than +30 mV or smaller than −30 mV.

Antibacterial activities of CuO/Cu_2_O NPs against three Gram-positive (G+) and five Gram-negative (G−) foodborne pathogens are shown in Figure 2. In G+ bacteria, the antimicrobial effect on *B. cereus* (ATCC 11778; 67.3%) was lower than on *L. monocytogenes* EGD (100.0%). The growth inhibition rates of CuO/Cu_2_O NPs on G− bacteria ranged from 82.8% to 100.0%. Figure 3 and Figure 4 showed colony counts of various foodborne pathogens after the NP treatment. Thirty microliters of each serial dilution were dropped on TSA plates. In the control sample, *S.* Typhimurium DT7 growth was observed at 10^−6^-dilution (mean CFU/mL: 1.1 × 10^8^), but after treatment with CuO/Cu_2_O NPs, colonies of *S*. Typhimurium DT7 were first observed at 10^−3^-dilution (mean CFU/mL: 1.0 × 10^5^), demonstrating a great reduction in the number of bacteria (Figure 3). Most significantly, CuO/Cu_2_O NPs resulted in the complete killing of *E. coli* O157:H7, *V. parahaemolyticus,* and *L. monocytogenes* cells (Figure 3 and Figure 4). FESEM images showed that the control cells were intact and rod-shaped with a smooth exterior (Figure 5 and Figure 6). In contrast, the bacterial cells exposed to CuO/Cu_2_O NPs exhibited a ruptured cell wall and complete lysis. The micrographic images demonstrated that disruption of the CuO/Cu_2_O NP-treated bacterial cells resulted in the formation of a large amorphous mass.

## 4. Discussion

NPs are increasingly used to target bacteria as an alternative to antimicrobial agents [25,26]. The antibacterial activity of CuO/Cu_2_O NPs synthesized by PLAL against G+ (*L. monocytogenes* EGD, *B. cereus* ATCC 11778) and -negative (*S*. Typhimurium DT7, *E. coli* O157:H7, *S. sonnei* ATCC 9290, *Y. enterocolitica* ATCC 27729, *V. parahaemolyticus* ATCC 49398) foodborne pathogenic bacteria was assessed by plate count method and FESEM. Overall, CuO/Cu_2_O NPs showed strong antimicrobial properties against both G+ and G− foodborne pathogens by exhibiting a “*contact-killing*” mechanism. This mechanism has already been observed by Gonçalves et al. [19] against clinical pathogens *S. aureus* (G+), *K. pneumoniae* (G−), and *P. aeruginosa* (G−) and by Meghana et al. [27] against *E. coli* (G−). This mechanism seems to be different from the “*drilling-killing*” mechanism of selenium nanoparticles [28,29] and the “*wrapping-killing*” mechanism of bismuth oxide flakes [30].

Colonies of *S.* Typhimurium (G−) showed uniform sizes without the presence of CuO/Cu_2_O NPs, while with the presence of CuO/Cu_2_O NPs, they showed various sizes; including small colonies. Small colony variants (SCVs) are slow-growing cells forming small-sized colonies that can revert to wild-type cells or even remain stable. This is due to environmental stresses such as antimicrobial agents, reactive oxygen species (ROS), low pH, and limited nutrition [31]. To the best of our knowledge, this is the first observation to report SCVs following the nanoparticle treatment against foodborne pathogens. These results indicate that CuO/Cu_2_O NPs may influence the bacterial phenotype and inhibit bacterial growth.

The antibacterial activity of copper NPs has been studied against various microorganisms such as *E. coli* (G−), *V. cholera* (G−), *P. aeruginosa* (G−), *P. Vulgaris* (G−), *L. monocytogenes* (G+), and *S. aureus* (G+) [32,33,34,35,36,37,38]. Yoon et al. confirmed that 33.5 and 28.2 ppm of CuO NPs showed a growth reduction of 90% for *E. coli* (G−) and *B. subtilis* (G+) [33]. Antibacterial tests of CuO NPs exhibited approximately a 99.8% reduction in bacterial growth at 3750 and 2500 ppm for *E. coli* (G−) and *S. aureus* (G+) strains [34]. CuO NPs made by thermal plasma technology had inhibitory activity against *E. coli* (G−) and *S. aureus* (G+), with minimum bactericidal concentrations ranging from 100 ppm to 5000 ppm [39]. Betancourt-Galindo et al. [40] found that 1600 and 3200 ppm were required for complete inhibition of *P. aeruginosa* (G−) and *S. aureus* (G+), respectively. Compared to the previous reports, our CuO/Cu_2_O NPs showed excellent bacterial growth inhibition at lower concentrations (3 ppm).

Because G+ bacteria have a thicker layer of peptidoglycan cell wall than G− bacteria, NPs have been known to penetrate the cell membrane of G− bacteria better [41]. A more significant bactericidal effect of copper oxide NPs was found on the growth of G− bacteria (*E. coli, P. aeruginosa*) than G+ bacteria (*B. subtilis*, *S. aureus*) [42,43]. On the contrary, Azam et al. and Tiwari et al. reported that CuO NPs exhibited greater antimicrobial activity against G+ *B. subtilis* and *S. aureus* than G− *E. coli* and *P. aeruginosa* [44,45]. In our study, CuO/Cu_2_O NPs not only significantly reduced the growth of G− bacteria, but also completely killed *L. monocytogenes* EGD (G+). Therefore, further studies on the antimicrobial mechanism by the copper oxide NPs are required to confirm these phenomena.

The antimicrobial activity of metal-based NPs is size-dependent, with small spherical NPs showing better bacterial killing activity than larger ones. Due to their large surface-to-volume ratio, the NPs actively contacted with cell membranes, disrupting the respiratory system resulting in bacterial death [46,47]. Chakra et al. [48] demonstrated that the smaller the size of copper-based NPs, the higher the penetrating power, killing the bacterial cells. An additional report indicated that the antibacterial activity against *E. coli* (G−) increased as the particle size of the NPs decreased from the micron size to the nanometer size [49]. In this study, FESEM images showed detailed interactions between CuO/Cu_2_O NPs and the bacterial cells. Even though CuO/Cu_2_O NPs were varied in size, only small-sized and spherical NPs less than 100 nm were attached to the whole surface of the bacterial cell membrane. The bacteria treated with CuO/Cu_2_O NPs showed cell wall and membrane disruption, with leakage of intracellular material, and complete cell lysis consistent with SEM observations [50].

## 5. Conclusions

Nanoparticles are currently contributing substantially to the development of innovative solutions in the agricultural and food sectors. Several factors can have an impact on the antibacterial effect of NPs against microorganisms, including the size, shape, composition, and concentration of the NPs. In this paper, CuO/Cu_2_O NPs have been produced by PLAL. The current limitation of the PLAL synthesis protocol is the productivity rate, which is limited to a few mg/h in this case. But, the main advantage of the method is the cleanliness of the surface of the CuO/Cu_2_O NPs being produced. Indeed, they do not have surfactants attached to their surface to induce separation between them as their separation is ensured by electrostatic repulsion. The close interaction of CuO/Cu_2_O NPs with the bacteria, called the “*contact-killing*” mechanism, completely lysed the cells. More work is in progress to test those NPs against clinical pathogens.

## Figures and Tables

**Figure 1 nanomaterials-13-02206-f001:**
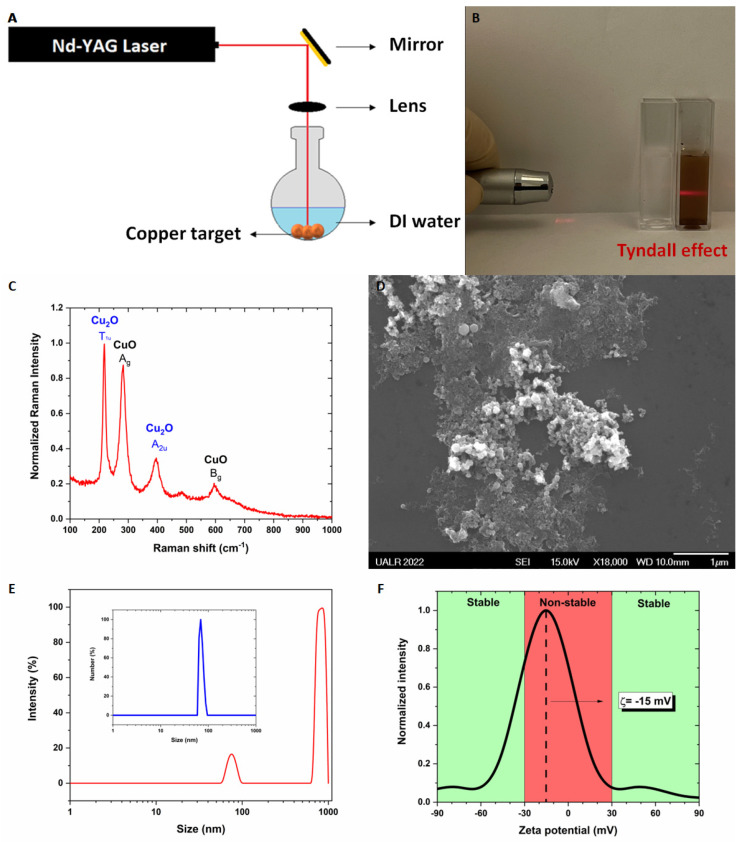
Physico-chemical characterization of the CuO/Cu_2_O NPs. (**A**) Sketch of the PLAL set-up. (**B**) Tyndall effect observed in the colloid synthesized by PLAL. The left container contains DI water only and serves as a reference while the right container contains the CuO/Cu_2_O NPs synthesized by PLAL. (**C**) Raman spectra of the colloid. (**D**) Representative SEM image of the CuO/Cu_2_O NPs. (**E**) Size distribution (intensity versus size) of CuO/Cu_2_O NPs measured by DLS. Inset. Number of particles versus size. (**F**) Zeta potential of CuO/Cu_2_O NPs.

**Figure 2 nanomaterials-13-02206-f002:**
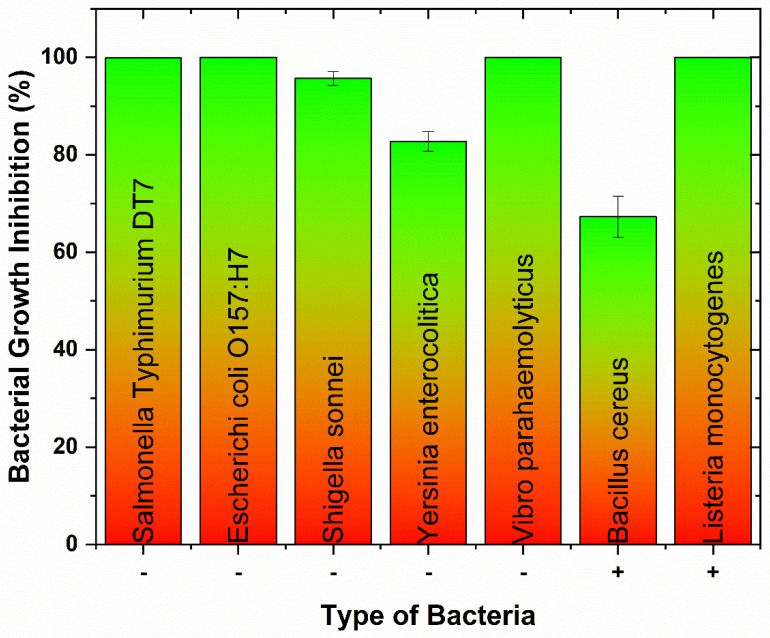
Growth inhibition of Gram-negative and -positive foodborne pathogens by CuO/Cu_2_O NPs. -: Gram-negative bacteria, +: Gram-positive bacteria.

**Figure 3 nanomaterials-13-02206-f003:**
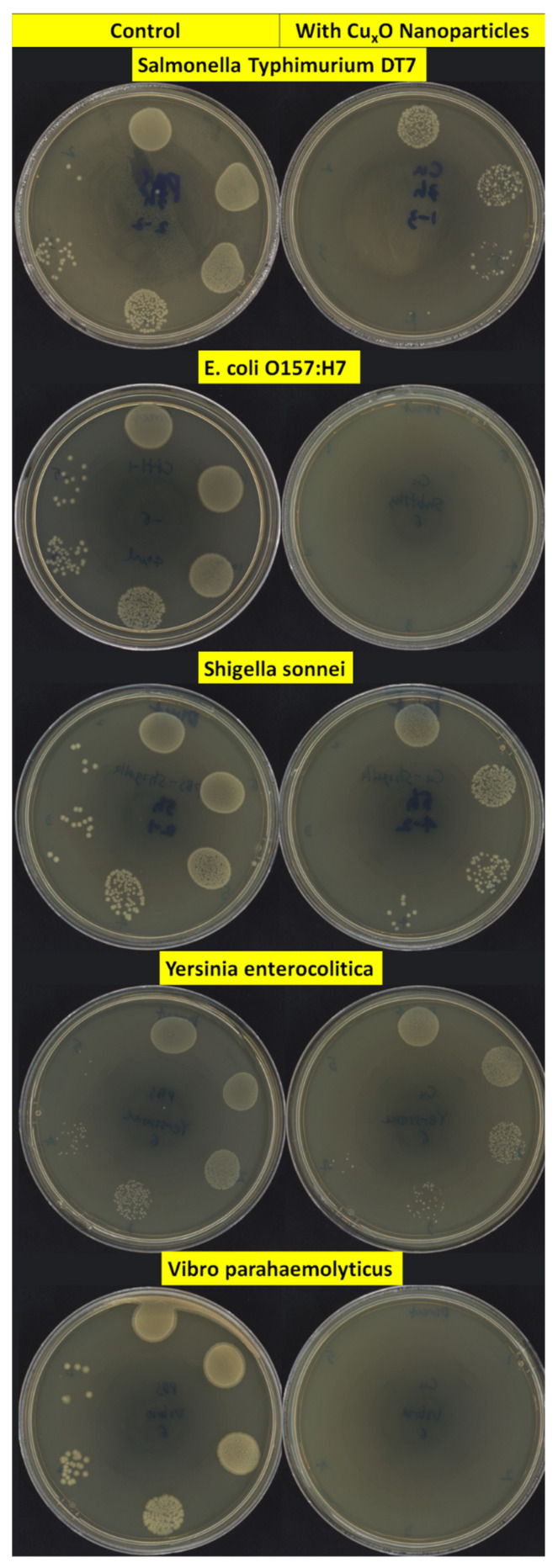
Antibacterial assay of CuO/Cu_2_O NPs against Gram-negative foodborne pathogen by agar plate count. **Left**—untreated (control), **Right**—CuO/Cu_2_O-treated.

**Figure 4 nanomaterials-13-02206-f004:**
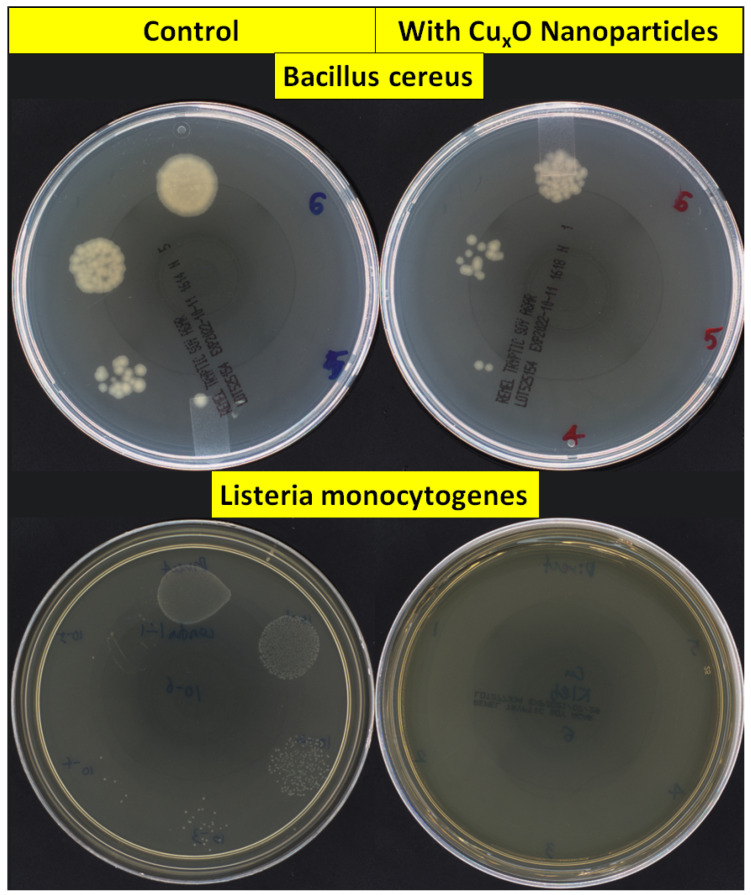
Antibacterial assay of CuO/Cu_2_O NPs against Gram-positive foodborne pathogen by agar plate count. **Left**—untreated (control), **Right**—CuO/Cu_2_O-treated.

**Figure 5 nanomaterials-13-02206-f005:**
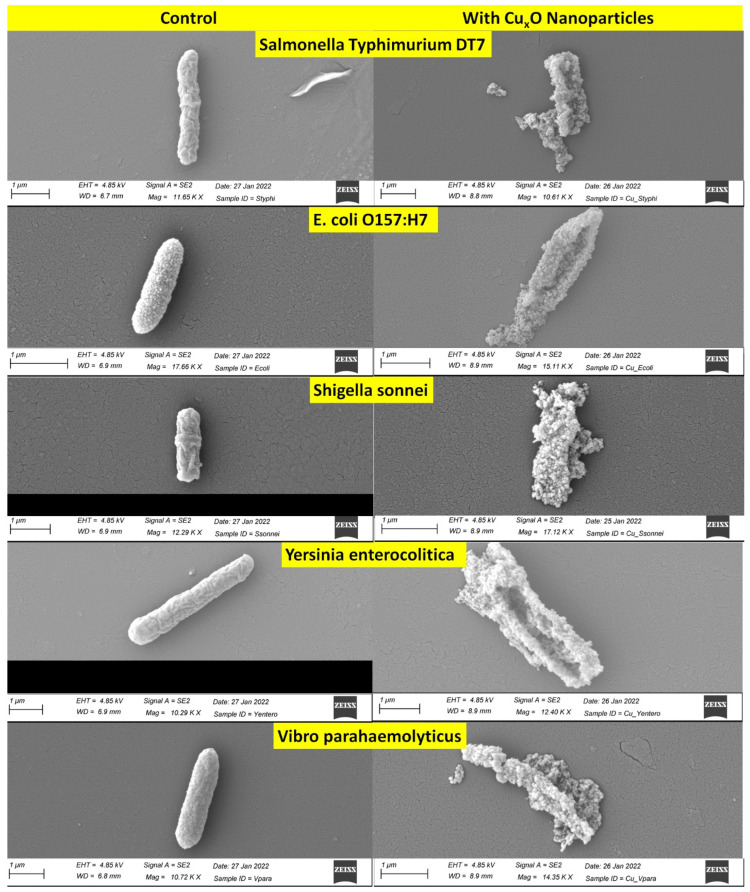
FESEM images of Gram-negative foodborne pathogens. The scale bar in all the images corresponds to 1 μm. **Left** images are untreated (control) and **right** images are CuO/Cu_2_O-treated.

**Figure 6 nanomaterials-13-02206-f006:**
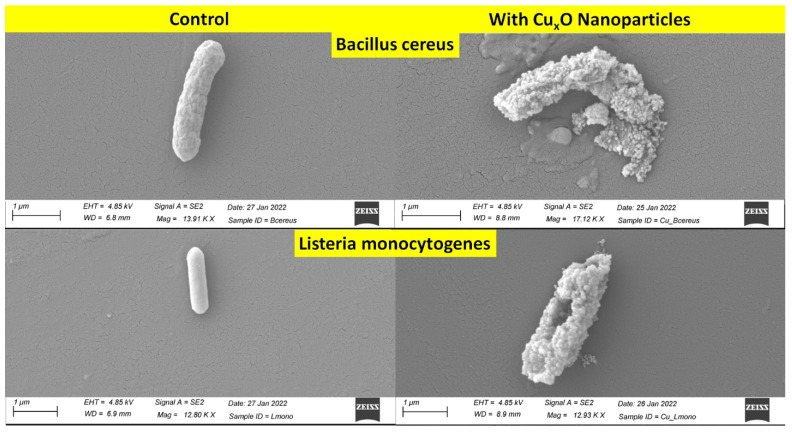
FESEM images of Gram-positive foodborne pathogens. The scale bar in all the images corresponds to 1 μm. **Left** images are untreated (control) and **right** images are CuO/Cu_2_O-treated.

## Data Availability

Data will be made available by the authors upon reasonable request.

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
