# Peer review of "Synthesis of Antibacterial Copper Oxide Nanoparticles by Pulsed Laser Ablation in Liquids: Potential Application against Foodborne Pathogens"

_nanomaterials, 2023, doi:10.3390/nano13152206_

Round 1
Reviewer 1 Report
The manuscript entitled "Synthesis of antibacterial copper oxide nanoparticles by pulsed laser ablation in liquids: potential application against food-borne pathogens", and written by Tina Hesabizadeh, Kidon Sung, Miseon Park, Steven Foley, Angel Paredes, Stephen Blisset and Gregory Guisbiers, provides some insights on the laser-mediated synthesis of CuO/Cu2O NPs and their use as antibacterial agents against food-borne pathogens. However, the manuscript seems incomplete, and it is not clear what is the significant contribution of the current manuscript to the field's state-of-the-art.
The manuscript could be considered for publication after clarifying and addressing the following major and minor comments.
Major comments:
1- Although the authors state the following: "Indeed, CuO/Cu2O NPs interact with sulfhydryl (-SH) groups and produce reactive oxygen species (ROS) that ultimately cause irreversible damage to the pathogen cells." They do not provide any data supporting this statement.
2- Why do the authors employ ethanol to dehydrate the bacteria? Ethanol is known to disrupt the cell membrane of most of these bacteria. Therefore its use must be clearly justified; otherwise, it can lead to a misleading interpretation of FESEM results.
3- Why do the authors not employ any stabilization strategy? Using monovalent salts like NaCl could help increase the NPs' electrical double layer and their electrostatic repulsion without compromising their ligand-free surface. In that way, the NPs antibacterial performance would be maximized.
4- Figures 5 and 6 look great. However, it would be better to have more general pictures to show the same phenomenon happening to most bacteria in the corresponding colonies.
5- The statement: "Because G+ bacteria have a thicker layer of peptidoglycan cell wall than G- bacteria, 182 NPs have known to penetrate the cell membrane of G- bacteria better" is only valid for NPs with relatively small sizes in general below or equal to 10 nm. However, the authors are reporting hydrodynamic sizes of ~100 nm and ~1 micron.
6- Following the previous comment, why did the authors not separate big from small-sized NPs and treat bacteria with both different types of NPs? As the authors correctly indicate, the NPs size reduction dramatically impacts their antibacterial activity.
7- Please provide the NPs zeta potential value to support the following conclusion: "Their antibacterial property may be due to the opposite electrical charges making them adhere tightly to the bacteria,"
8- There is no data supporting the statement that there is a strong reduction reaction at the bacterial cell walls. The FESEM data only shows disruption of the cell membrane integrity, not any sign of a reduction mechanism.
Minor comments:
1- Please provide the reasoning behind selecting the employed experimental conditions, including laser synthesis parameters and antibacterial test conditions.
Author Response
Please see our responses in the attached file. Thank you.

Reviewer 2 Report
In this article, the authors present their investigation on the synthesis of spherical copper oxide nanoparticles (CuO/Cu2O NPs) through pulsed laser ablation in liquids (PLAL). The size distribution and concentration of the NPs were determined using dynamic light scattering (DLS) and atomic emission spectroscopy (AES), respectively. Raman spectroscopy was employed to identify the phases of copper oxide. Furthermore, the authors evaluate the antibacterial activity of the CuO/Cu2O NPs against various foodborne pathogens. Notably, the NPs demonstrated potent antimicrobial effects, effectively eradicating the majority of bacteria after a 5-hour incubation period at 25 °C when tested at a concentration of 3 parts per million (ppm). To confirm the destructive impact of the CuO/Cu2O NPs on the bacterial cell wall, field emission scanning electron microscopy (FESEM) was utilized.
While the paper provides comprehensive information and is well-written, with clear concepts, there is a significant absence of detailed experimental procedures that would allow for replication by other research groups. Moreover, the inclusion of supplementary experiments would greatly enhance the paper. Considering the esteemed quality standards of the high-impact Nanomaterial journal, I recommend significant revisions before publication. Below, I have outlined some specific suggestions:
1) The introduction lacks depth, particularly regarding referencing previous works conducted by other researchers on the synthesis of copper oxide nanoparticles nanoparticles using PLAL, and even used as biocidal elements. It is essential to include appropriate citations to acknowledge and build upon the existing body of research in this field.
2) The authors need to provide more information on the laser parameters required for nanoparticle synthesis. Details such as the beam size and fluence of the experiment are missing.
3) As a curiosity, Figure 1D illustrates the size distribution in the intensity measured with DLS. In number the micrometer size nanoparticles would be less than the nanometer size. However, it is unclear whether any procedure was performed to remove particles around 1 micron from the sample. Can the authors provide clarification on this matter?
4) My major critique is the lack of experiments providing detailed information about the biocidal properties of the copper oxide nanoparticles. It is crucial to note that the outcome of the bacterial growth inhibition percentage is influenced by several factors, including the concentration of the antimicrobial agent and the duration of exposure. To provide a more comprehensive understanding of the biocidal effect of the synthesized nanoparticles, I recommend conducting additional experiments. Specifically, it is important to calculate the Minimum Inhibitory Concentration (MIC) and the Minimum Bactericidal Concentration (MBC) for each bacterium. These values can help to evaluate the effectiveness of the NPs against different strains of bacteria. Additionally, assessing the cytotoxicity profile of the nanoparticles is crucial for considering their real-world use. Conducting time-dependent cytotoxicity tests using assays such as MTT (3-(4,5-dimethylthiazol-2-yl)-2,5-diphenyltetrazolium bromide) or Alamar Blue can provide insights into the effects of the nanoparticles on cell viability over time. This information is valuable in assessing the potential risks and safety considerations associated with the use of these nanoparticles in various applications.
Author Response
Please, see our responses in the attached file. Thank you.

Reviewer 3 Report
The present work concerns the synthesis of copper oxide nanoparticles by pulsed laser ablation. The physical-chemical characterization of nanoparticles is presented as well as the antimicrobial investigation against foodborne pathogens is shown. Since the demand of new strategy and tools for the prevention of foodborne illness is still growing up, the present study could be suitable for the publication in the Special Issue “Innovative Biomedical Applications of Laser-Generated Colloids” of Nanomaterials journal. However, minor revision is required to improve the current version of the manuscript.

Author Response

(The authors gave the same response as above.)

Round 2
Reviewer 1 Report
The authors adressed the major and minor comments appropriately. Therefore, now I can recommend the current manuscript for its publication.